# Real-World Study on Effectiveness of Molnupiravir and Nirmatrelvir–Ritonavir in Unvaccinated Patients with Chronic Respiratory Diseases with Confirmed SARS-CoV-2 Infection Managed in Out-Patient Setting

**DOI:** 10.3390/v15030610

**Published:** 2023-02-23

**Authors:** Wang Chun Kwok, Man Fung Tsoi, Sze Him Isaac Leung, Chung Ki Tsui, Terence Chi Chun Tam, James Chung Man Ho, David Chi Leung Lam, Mary Sau Man Ip, Pak Leung Ho

**Affiliations:** 1Department of Medicine, The University of Hong Kong, Queen Mary Hospital, Hong Kong, China; 2Centre for Epidemiology Versus Arthritis, The University of Manchester, Manchester M139PT, UK; 3Department of Statistics, The Chinese University of Hong Kong, Hong Kong, China; 4Department of Medicine, Queen Mary Hospital, Hong Kong, China; 5Department of Microbiology and Carol Yu Centre for Infection, The University of Hong Kong, Queen Mary Hospital, Hong Kong, China

**Keywords:** COVID-19, COPD, asthma, bronchiectasis, molnupiravir, nirmatrelvir–ritonavir

## Abstract

While molnupiravir (MOV) and nirmatrelvir–ritonavir (NMV-r) were developed for treatment of mild to moderate COVID-19 infection, there has been a lack of data on the efficacy among unvaccinated adult patients with chronic respiratory diseases, including asthma, chronic obstructive pulmonary disease (COPD) and bronchiectasis. A territory-wide retrospective cohort study was conducted in Hong Kong to investigate the efficacy of MOV and NMV-r against severe outcomes of COVID-19 in unvaccinated adult patients with chronic respiratory diseases. A total of 3267 patients were included. NMV-r was effective in preventing respiratory failure (66.6%; 95% CI, 25.6–85.0%, *p =* 0.007), severe respiratory failure (77.0%; 95% CI, 6.9–94.3%, *p =* 0.039) with statistical significance, and COVID-19 related hospitalization (43.9%; 95% CI, −1.7–69.0%, *p =* 0.057) and in-hospital mortality (62.7%; 95% CI, −0.6–86.2, *p =* 0.051) with borderline statistical significance. MOV was effective in preventing COVID-19 related severe respiratory failure (48.2%; 95% CI 0.5–73.0, *p =* 0.048) and in-hospital mortality (58.3%; 95% CI 22.9–77.4, *p =* 0.005) but not hospitalization (*p =* 0.16) and respiratory failure (*p =* 0.10). In summary, both NMV-r and MOV are effective for reducing severe outcomes in unvaccinated COVID-19 patients with chronic respiratory diseases.

## 1. Introduction

The development of oral antiviral drugs significantly changed the paradigm of the treatment of coronavirus disease 2019 (COVID-19). Molnupiravir (MOV) and nirmatrelvir–ritonavir (NMV-r) have been shown in clinical trials to reduce hospitalization and mortality [1,2,3]. The MOVe-OUT study suggests that early administration of MOV to non-hospitalized patients with mild-to-moderate COVID-19 accelerates viral clearance and reduces the relative risk of hospitalization or mortality by 30% [1]. In the EPIC-HR trial, NMV-r reduced the rates of hospitalization and mortality by 89% when the drug was initiated within 3 days of symptom onset, and by 88% if initiated within 5 days of symptom onset, in non-hospitalized patients with mild-to-moderate COVID-19 who were at risk of progression to severe disease [4].

Real-world studies have also suggested the benefits of MOV and NMV-r. Early initiation of oral antivirals was associated with reduced risks of mortality and in-hospital disease progression among non-hospitalized patients with COVID-19 with additional benefits from NMV-r in terms of reducing the risks of hospitalization [5]. Initiation of MOV and NMV-r in hospitalized patients not requiring oxygen therapy on admission was also shown to have substantial clinical benefit in terms of all-cause mortality, composite disease progression outcome and time to achieving a low viral burden [6]. In Hong Kong, MOV and NMV-r prescriptions in high-risk patients with mild-to-moderate COVID-19 were also associated with significant cost savings [7]. In another real-world study conducted in Hong Kong, the use of NMV-r but not MOV was associated with a reduced risk of hospitalization in non-hospitalized COVID-19 patients [8].

While both MOV and NMV-r have been shown to be effective in high-risk patients, there is limited evidence in patients with chronic respiratory diseases, which, including asthma, chronic obstructive pulmonary disease (COPD) and bronchiectasis, are at particularly increased risks of adverse COVID-19 outcomes [9,10,11,12,13,14]. Asthma was reported as a risk factor for severe COVID-19 infection with respiratory and systemic complications [9] and mortality [15]. Patients with COPD were reported to have increased risks of hospitalization, intensive care unit (ICU) admission and mortality [16]. COVID-19 in patients with bronchiectasis was reported to be associated with higher likelihood to receive supplemental oxygen and extracorporeal membrane oxygenation (ECMO) and higher mortality than those without bronchiectasis [17]. In this study, we aim at assessing the clinical efficacy of MOV and NMV-r among unvaccinated patients with chronic respiratory disease, including asthma, COPD and bronchiectasis, who were managed in an out-patient setting.

## 2. Materials and Methods

### 2.1. Study Design and Data Sources

This was a territory-wide retrospective cohort study on the efficacy of MOV and NMV-r among adult unvaccinated patients with chronic respiratory diseases, including asthma, COPD and bronchiectasis, who were managed in an out-patient setting. Patients were identified from the Clinical Data Analysis and Reporting System (CDARS) of the Hospital Authority (HA). CDARS is an electronic healthcare database managed by the HA that covers 90% of healthcare services of Hong Kong, as well as managing the vast majority of patients with COVID-19 in Hong Kong. HA is the statutory body that operates all public hospitals and clinics in Hong Kong, including 23 designated clinics for outpatient COVID-19 treatments, and it was responsible for all COVID-19 related treatments during the study period. In Hong Kong, public health care service is provided by HA, which manages 43 hospitals and institutions and 122 outpatient clinics, managing more than 90% of the patients in Hong Kong. HA is also responsible for coordinating outpatient and inpatient care of COVID-19 patients. The majority of patients with COVID-19 are managed in the clinics or hospitals under HA. Patients with COVID-19 are managed in an out-patient setting in a designated clinic if they are clinically stable. Patients who have moderate to severe disease or multiple medical co-morbidities are admitted to an airborne infection isolation room (AIIR) with standard, contact, droplet and airborne precautions [18]. Early COVID-19 treatments are offered to inpatients with increased risk of severe disease. The HA Central Committee on Infectious Diseases and Emergency Response (CCIDER) issues updated an Interim Recommendation on Clinical Management of Adult Cases with Coronavirus Disease 2019 regularly to guide the treatment pathway for COVID-19 cases. MOV was available in Hong Kong for prescription from 26 February 2022 onwards, and NMV-r from 16 March 2022 onwards. The following are considered to increase the risk of severe disease: diabetes mellitus, being obese with body mass index more than 30 kg/m^2^, age above 60 years, immunocompromised state, having underlying chronic illnesses and having incomplete COVID-19 vaccination. For patients who have mild symptoms but are at risk of disease progression and are within the early onset of disease (within 5 days), NMV-r or MOV is started, with the former preferred if not contraindicated.

Cases with COVID-19 were identified using the International Classification of Diseases, Ninth Revision (ICD-9) code of 519.8 for the period between 26 February 2022 and 26 August 2022. Patients with asthma, COPD and bronchiectasis were identified using the ICD-9 codes of 493, 496 and 494, respectively. To allow time for development of the measured outcomes, data for the outpatient cohort were censored on 26 September 2022. The vaccination record of the patients with chronic respiratory diseases and COVID-19 were identified from CDARS, with the subjects who did not have a vaccination record being excluded. Patients who had past COVID-19 before the study start date were also excluded. Our exposure of interest was MOV and NMV-r prescriptions among COVID-19 patients who were unvaccinated patients with chronic respiratory diseases who were managed in a designated COVID-19 clinic. The standard course of both antivirals was 5 days [18].

The study was approved by the Institutional Review Board of the University of Hong Kong and Hospital Authority Hong Kong West Cluster (UW 22-739).

### 2.2. Outcomes

The primary outcomes were the incidences of COVID-19 related hospitalization, respiratory failure, severe respiratory failure and mortality. COVID-19 related hospitalization was defined by admission to an acute medical ward for management of COVID-19 infection for more than 24 h. COVID-19 related respiratory failure was defined as desaturation with saturation of oxygen below 90% within 14 days of confirmed COVID-19 infection. COVID-19 related severe respiratory failure was defined as the need for non-invasive ventilation or invasive mechanical ventilation within 14 days of confirmed COVID-19 infection [19]. COVID-19 related mortality was defined as inpatient death during COVID-19 related hospitalization.

### 2.3. Statistical Analysis

The demographic and clinical data were described in actual frequency and mean ± standard deviation (SD) or median (interquartile range). Baseline demographic and clinical data were compared between the three groups (no antiviral, MOV and NMV-r) with a one-way ANOVA test for continuous variables and Pearson’s χ^2^ test for categorical variables. The MOV group and NMV-r group were separately compared with the no antiviral group using the χ^2^ test and unpaired t test for categorical and continuous variables, respectively. To identify whether antivirals were associated with protection from COVID-19 related respiratory failure, COVID-19 related severe respiratory failure and COVID-19 related mortality, which were binary outcomes counted as incidence, univariate log-binomial regression analyses were performed. Poisson or negative binomial regressions were not used, as they are used for modelling count data but not binary data outcomes. Multiple log-binomial regression modeling was used to account for potential confounds, including age, gender, underlying respiratory disease (asthma, COPD or bronchiectasis), baseline estimated glomerular filtrate rate (eGFR) as calculated using the modification of diet in renal disease (MDRD) equation and Charlson comorbidity index (CCI) via a multivariate analysis model. For age, eGFR and CCI, they were entered as continuous variables in multivariate log-binomial regression. The statistical significances were determined at the level of *p* < 0.05. Risk ratio (RR) and 95% confidence intervals (CI) for the outcomes in the MOV, NMV-r and no treatment groups were calculated using log-binomial regression. Sensitivity analysis was performed using logistic regression. Anti-viral effectiveness was calculated as (1 − adjusted RR [aRR)) × 100. All the statistical analyses were performed using R version 4.2.2.

## 3. Results

### 3.1. Patients’ Characteristics

A total of 3267 adult patients with asthma, COPD and bronchiectasis were identified from the clinic cohort. Among the 3267 patients in the cohort, 2387 (73.1%) patients did not receive anti-viral treatment, 302 (9.2%) received NMV-r and 578 (17.7%) patients received MOV. Among the 3267 patients included in the analysis, there were 2341 (71.7%) male patients, with a mean age of 79.4 ± 14.0 years. There were 2065 (63.2%) patients with COPD, 685 (21.0%) with asthma and 517 (15.8%) with bronchiectasis. The median baseline CCI was 1 (interquartile range = 1–2). Patients treated with NMV-r were younger and less likely to be on concomitant medications that may interact with NMV-r and also had better renal function, as measured by eGFR. The baseline demographics of the patients are listed in Table 1 and Appendix A. The monthly proportion of COVID-19 patients prescribed with NMV-r and MOV in our cohort is illustrated in Figure 1.

#### 3.1.1. COVID-19 Related Hospitalization

Among the included patients, 8.8% (287/3267) of them required hospitalization for COVID-19 infection. Patients who did not receive ant-viral treatment (228/2387, 9.6%) were more likely to require hospitalization for COVID-19 infection than those who received NMV-r (5.0%, 15/302) and MOV (7.6%, 10/578) (Table 2 and Appendix A). The unadjusted RR by univariate log-binomial regression was 0.52 (95% CI = 0.31–0.87, *p* = 0.01) for NMV-r and 0.80 (95% CI = 0.59–1.09, *p* = 0.15) for MOV. The aRR after accounting for potential confounds including age, gender, underlying respiratory disease, baseline eGFR and CCI was 0.56 (95% CI = 0.31–1.02, *p* = 0.057) for NMV-r and 0.782 (95% CI = 0.55–1.11, *p* = 0.161) for MOV.

#### 3.1.2. COVID-19 Related Respiratory Failure

Overall, 7.9% (259/3267) had COVID-19 related respiratory failure (Table 2 and Appendix A). Patients who did not receive ant-viral treatment (8.8%, 209/2387) were more likely to have COVID-19 related respiratory failure than those who received NMV-r (3.3%, 10/302) and MOV (6.9%, 40/578). However, the difference was only statistically significant for NMV-r. The unadjusted RR by univariate log-binomial regression was 0.378 (95% CI = 0.20–0.71, *p* < 0.01) for NMV-r and 0.79 (95% = 0.57–1.10, *p =* 0.16) for MOV. The aRR after accounting for potential confounds including age, gender, underlying respiratory disease baseline eGFR and CCI were 0.334 (95% CI = 0.15–0.74, *p* < 0.01) for NMV-r and 0.74 (95% CI = 0.51–1.06, *p =* 0.10) for MOV.

#### 3.1.3. COVID-19 Related Severe Respiratory Failure

Overall, 3.4% (110/3267) patients had COVID-19 related severe respiratory failure (Table 2 and Appendix A). Patients who did not receive ant-viral treatment (3.8%, 91/2378) were more likely to have COVID-19 related severe respiratory failure than those who received NMV-r (1.0%, 3/302) and MOV (2.8%, 16/578). The RR by univariate log-binomial regression was 0.26 (95% CI = 0.08–0.82, *p =* 0.02) for NMV-r and 0.73 (95% = 0.43–1.23, *p* = 0.23) for MOV. The aRR after accounting for potential confounds including age, gender, underlying respiratory disease baseline eGFR and CCI was 0.23 (95% CI = 0.06–0.93, *p* = 0.04) for NMV-r and 0.52 (95% CI = 0.27–0.99, *p* = 0.05) for MOV.

#### 3.1.4. COVID-19 Related Mortality

Overall, 4.3% (139/3267) patients died (Table 2 and Appendix A). Patients who did not receive ant-viral treatment (5.0%, 120/2378) were more likely to die than those who received NMV-r (1.3%, 4/302) and MOV (2.6%, 15/578). The RR by univariate log-binomial regression was 0.26 (95% CI = 0.10–0.71, *p* < 0.01) for NMV-r and 0.52 (95% = 0.30–0.88, *p* = 0.01) for MOV. The aRR after accounting for potential confounds including age, gender, underlying respiratory disease, baseline eGFR and CCI was 0.37 (95% CI = 0.14–1.01, *p* = 0.05) for NMV-r and 0.42 (95% CI = 0.23–0.77, *p* < 0.01) for MOV.

### 3.2. Sensitivty Analysis

The sensitivity analyses using logistic regression produced results similar to the primary analyses (Appendix A).

## 4. Discussion

Our study contains the first real world data to show that both NMV-r and MOV are effective in preventing COVID-19 related complications in unvaccinated patients with chronic respiratory diseases. NMV-r can prevent COVID-19 related respiratory failure and COVID-19 related severe respiratory failure with statistical significance. The results on COVID-19 related hospitalization and mortality upon hospitalization for COVID-19 infection also reached borderline significance. MOV was shown to prevent COVID-19 related severe respiratory failure and in-hospital mortality upon hospitalization for COVID-19 infection.

Since the launch of NMV-r and MOV, there has been a major shift in COVID-19 treatment paradigm. Both clinical trials and real-world study suggest the benefits of both oral antivirals in reducing hospitalization and mortality [1,2,3,4,5,6,7,8]. In a systemic review and meta-analysis, treatment with both NMV-r and MOV were associated with a significantly lower risk of COVID-19 related hospitalization or death compared with the placebo, with the benefits being more pronounced for NMV-r than MOV [20]. A real-world study in Italy suggested both NMV-r and MOV to be effective, while patients treated with NMV-r had lower risk of hospitalizations and earlier recovery, though the age, body mass index and underlying co-morbidities differed in the NMV-r and MOV groups, as in our study as well as other real-world studies [21]. In a recent retrospective, observational, nationwide propensity-matched analysis comparing outcomes of US veterans 65 years and older with mild-to-moderate COVID-19, the composite primary outcome of admission or death within 30 days of diagnosis was reached less often in those receiving either MOV or NMV-r versus those that received no antiviral, while NMV-r was more effective than MOV in both outcomes [22]. The composite outcome of death or hospitalization was shown to be lower in NMV-r treated patients than MOV treated subjects in another study, but the difference in the degree of underlying co-morbidity could contribute to the difference [23]. Viral rebound was also shown to be uncommon for patients treated with both NMV-r and MOV [24].

In Hong Kong, the HA CCIDER issues an updated Interim Recommendation on Clinical Management of Adult Cases with COVID-19 regularly to guide the treatment pathway for COVID-19 cases. Patients with underlying chronic illnesses who have had incomplete COVID-19 vaccination are considered to have high-risk for severe COVID-19 outcomes. NMV-r and MOV can be initiated in patients who have mild symptoms but are at risk of disease progression within 5 days of symptom onset, with the former preferred if not contraindicated. While studies have suggested the superiority of NMV-r over MOV in managing mild to moderate COVID-19 given its superior efficacy, MOV is more often used in patients who have contraindications to NMV-r, including patients with severe renal impairment (eGFR < 30 mL/min) and those who are on drugs that have major drug–drug interactions. Real-world studies have been conducted to compare the efficacy of antivirals in patients managed in out-patient settings. One real-world study conducted in Hong Kong suggested that MOV and NMV-r use were associated with lower risks of death and in-hospital disease progression [5]. However, only NMV-r use was additionally associated with a reduced risk of hospitalization. There was a consistent finding on reduced risks of mortality and hospitalization associated with early oral antiviral use among older patients in the same real-world study [5]. Another real-world study that was also conducted in Hong Kong between 16 February and 31 March 2022 found that the use of NMV-r but not MOV was associated with a reduced risk of hospitalization in COVID-19 patients [8]. Our study, which included the important subgroup of those who have chronic respiratory diseases, revealed similar findings: that NMV-r and MOV are effective in preventing adverse COVID-19 related outcomes. Most of the published literature on the efficacy of NMV-r and MOV has focused on the whole population [5,6,21]. There have also been studies on selected patient subgroups including all COVID-19 subjects at high risk of progression [23], elderly populations [22,25], hematology patients [26] and solid organ transplant recipients [27]. However, there has not been any dedicated study focusing on patients with chronic respiratory disease, who are prone to developing severe COVID-19. Apart from the propensity towards developing complications from COVID-19, any viral infection including COVID-19 can also trigger exacerbation of the underlying respiratory disease. Hence, it is important to assess anti-viral effectiveness specifically for patients with chronic respiratory diseases. These findings have significant implications in management of mild to moderate COVID-19 infection for unvaccinated patients with chronic respiratory diseases in an out-patient setting. While NMV-r is the preferred agent, MOV is still a reasonable choice given its efficacy in preventing COVID-19 related severe respiratory failure and in-hospital mortality upon hospitalization for COVID-19 infection. For patients with chronic respiratory diseases, it is common for them to have other co-morbidities and be on various medications that preclude the safe use of NMV-r. For these patients, if NMV-r is not considered to a suitable choice, MOV could be an alternative. For patients who were treated with MOV, early follow-up for disease progression would be warranted, as 7.6% of the patients treated with MOV needed hospitalization for COVID-19 infection.

There were more patients treated with MOV than NMV-r in our cohort. This could be explained by MOV being available earlier in Hong Kong than NMV-r. Figure 1 illustrates how MOV was more commonly prescribed in February and March 2022, as NMV-r was only available in the second half of March 2022. At the peak of the fifth COVID-19 wave in late February to early March 2022, only MOV was available in Hong Kong. The proportion of patients prescribed NMV-r exceeded those prescribed MOV after June 2022. However, by that time, the number of COVID-19 cases was dropping in Hong Kong. This could further explain why more MOV was prescribed than NMV-r overall within this cohort. At the same time, most of the patients within this cohort belonged to the elderly population. Although the treatment for chronic respiratory diseases does not affect the antiviral of choice per se, as the public health care system was overwhelmed in the fifth COVID-19 wave from February to April 2022, clinicians might have felt more comfortable prescribing MOV for elderly patients with comorbidities, as MOV has less drug–drug interaction than NMV-r and does not require dosage adjustment according to renal function. As such, clinicians may have been more willing to prescribe MOV from February to April 2022. A similar prescription pattern was also observed in other local real-world studies [5,6,7,8].

Although NMV-r and MOV have demonstrated clinical efficacy in unvaccinated patients with chronic respiratory diseases, the benefits of COVID-19 vaccination cannot be overemphasized. In Hong Kong, the uptake of COVID-19 vaccine among the elderly population was poor. By the end of February 2022, the proportions of the unvaccinated Hong Kong population aged over 60 years and aged over 80 years were 26% and 50%, respectively [28]. These were 64% and 59% of the subjects within our cohort above the age of 60 years and 80 years, respectively. The high percentage of elderly population within our cohort can explain the reason why we could identify a relatively large unvaccinated population. Numerous studies have clearly demonstrated the benefits of COVID-19 vaccination in preventing severe COVID-19, including in patients with chronic respiratory diseases [29,30,31,32,33,34]. Our group recently reported that BNT162b2 and CoronaVac vaccines are effective in preventing hospitalization for COVID-19 and respiratory failure complicating COVID-19 among patients with chronic respiratory diseases [30]. The cornerstone of successful control of COVID-19 includes vaccination and early treatment with antivirals for patients who at risk of disease progression.

Regarding the monoclonal antibody combination of tixagevimab–cilgavimab (Evusheld), due to limited availability, its use in public hospitals during the study period in Hong Kong was limited to patients with severe immunocompromising conditions including hematopoietic stem cell or solid organ (heart, liver and kidney) transplant recipients within 1 year of transplantation, lung transplant recipients, patients on B-cell depleting agents or Bruton tyrosine kinase inhibitors, patients with lymphoma or myeloma with active chemotherapy within 1 year, and patients with human immunodeficiency virus with CD4 count below 200 cells/mm^3^. Patients with chronic respiratory diseases were not entitled to receive monoclonal antibody prophylaxis according to the local guidelines and none of the patients within the cohort received monoclonal antibody combination of tixagevimab–cilgavimab [35]. One of the strengths of our study is that we selected a highly specialized population at high risk of severe COVID-19 infection. At the same time, the efficacy of the two antivirals was compared when Hong Kong had the fifth wave of COVID-19, which was contributed to by the Omicron variants. The results from our study concur with previous evidence for both NMV-r and MOV against the Omicron variants [5,6,27]. In this territory-wide study, we used public health-care databases that encompass the vast majority of the reported cases of confirmed COVID-19 infection during the observation period. Alongside the introduction of both oral antivirals in the public health-care system during this outbreak, their clinical effectiveness in unvaccinated patients with chronic respiratory diseases could be assessed in a real-world setting.

Despite this, there are some study limitations that have to be acknowledged. First, indication bias could not be eliminated in the prescription of oral antivirals, as reflected by the age and comorbidity differences among the three treatment groups. Residual confounding by indication could be present in the clinical decision to prescribe MOV versus NMV-r, because of the potential drug–drug interaction with various medications from NMV-r. Nevertheless, we adjusted for the possible confounds, including CCI and baseline eGFR, in multivariate analysis, which demonstrated consistent results. While we captured all reported patients in Hong Kong, we cannot rule out self-ascertainment bias, in that some of the patients might not be aware of having been infected, or did not report infection status to health authorities, thus becoming eligible to be offered the oral antivirals. There was a significant proportion of patients who were not prescribed antivirals in this study. The potential reasons could be delay in clinic attendance as well as patient refusal of treatment. For both NMV-r and MOV, they should be used within 5 days of symptom onset. Unfortunately, at the time of the study period, the public health care system that provided care to almost all the COVID-19 cases was overwhelmed, which could have caused delays in clinic appointment, making some patients fall out of the 5-day interval for antiviral treatment. Lack of confidence of new drugs when they were just launched in Hong Kong could also explain the low prescription rate. Another limitation of this study is that some of the clinical data including days from COVID-19 diagnosis/symptom onset and start of anti-viral treatment, symptomatic status and treatment were not available as they could not be retrieved from CDARS. As a non-randomized observational study, there was potential selection bias with possibility of residual confounding [36]. Yet, it may not be possible to examine the benefits of the antivirals in selected subgroups in randomized controlled trials (RCT), as they would be excluded due to the stringent exclusion criteria of RCTs. As such, non-randomized observational studies still have their value in assessing the benefits of antivirals for COVID-19, especially among subgroups that would not be included in RCTs.

## 5. Conclusions

NMV-r and MOV are effective in preventing severe COVID-19 related outcomes among unvaccinated patients with chronic respiratory diseases.

## Figures and Tables

**Figure 1 viruses-15-00610-f001:**
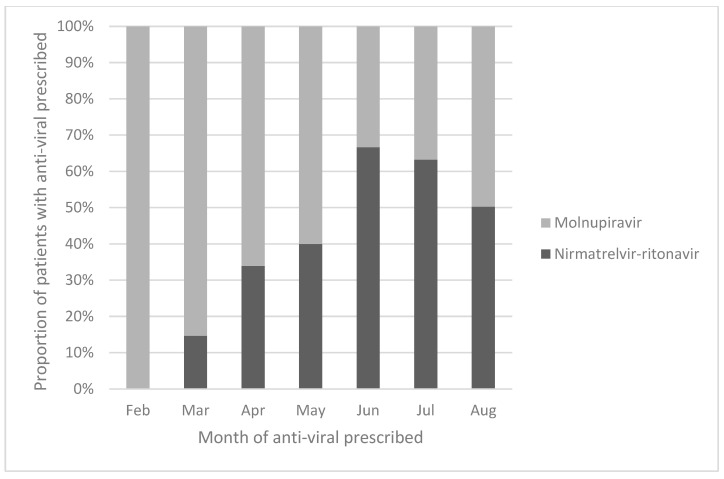
Proportion of patients prescribed nirmatrelvir–ritonavir and molnupiravir in this cohort.

**Table 1 viruses-15-00610-t001:** Demographic and clinical characteristics of included patients.

Variable ^a^	No Anti-Viral (*n* = 2387)	NMV-r (*n* = 302)	MOV (*n* = 578)	*p* Value ^c^
Age (years), median [IQR]	82 [73–89]	79 [71–87]	84 [75–90]	<0.001 *
Male	1762 (73.8%)	193 (63.9%)	386 (66.8%)	<0.001 *
Ethnic group				
Chinese	2336 (99.0%)	297 (98.3%)	576 (99.7%)	0.358
Caucasian	4 (0.2%)	0 (0%)	0 (0%)	
Southeast Asian	6 (0.3%)	1 (0.3%)	1 (0.2%)	
South Asian	9 (0.4%)	2 (0.7%)	1 (0.2%)	
Japanese	1 (0%)	1 (0.3%)	0 (0%)	
Others	4 (0.2%)	1 (0.3%)	0 (0%)	
Respiratory diseases				<0.001 *
Asthma	474 (19.9%)	73 (24.2%)	138 (23.9%)	
Bronchiectasis	358 (15.0%)	69 (22.8%)	90 (15.6%)	
COPD	1555 (65.1%)	160 (53.0%)	350 (60.6%)	
CCI, Median [IQR]	1 [1–2]	1 [0–1]	1 [1–2]	<0.001 *
Medication				
Statin	1040 (43.6%)	128 (42.4%)	308 (53.3%)	<0.001 *
Anti-epileptic	28 (1.2%)	2 (0.7%)	11 (1.9%)	0.229
DOACs Calcium channel blocker	191 (8.0%) 1294 (54.2%)	3 (1.0%) 146 (48.3%)	64 (11.1%) 351 (60.7%)	<0.001 * 0.001
Baseline blood test, median [IQR]				
Leucocyte count ^b^	6.08 [4.90–7.38]	6.10 [5.24–7.53]	6.00 [4.90–7.38]	0.146
Neutrophil count ^b^	3.86 [2.93–4.92]	3.93 [3.10–5.03]	3.90 [3.05–4.86]	0.473
Lymphocyte count ^b^	0.82 [0.50–1.32]	1.10 [0.63–1.52]	0.90 [0.53–1.40]	0.067
Eosinophil count ^b^	0.04 [0.00–0.14]	0.07 [0.00–0.15]	0.04 [0.00–0.14]	0.249
Estimated GFR, mL/min/1.73 m^2^	58.2 [40.5–77.0]	63.7 [51.7–81.8]	52.3 [34.7–74.4]	<0.001 *
ALT (unit/L)	12.0 [8.0–16.6]	13.3 [9.0–19.0]	12.0 [8–17.4]	0.107
COVID-related outcomes				
Hospitalization	228 (9.6%)	15 (5.0%)	44 (7.6%)	0.016 *
Respiratory failure	209 (8.8%)	10 (3.3%)	28 (6.9%)	0.073
Severe respiratory failure	91 (3.8%)	3 (1.0%)	16 (2.8%)	0.026 *
Mortality	120 (5.0%)	4 (1.3%)	15 (2.6%)	0.001 *
Length of stay (days), median [IQR]	9 [4–16]	9 [3–14]	9.5 [4–24.5]	0.138

ALT: Alanine aminotransferase; CCI: Charlson co-morbidity index; DOACs: direct-acting oral anticoagulants; GFR: glomerular filtration rate (mL/min/1.73 m^2^); IQR: interquartile range; MOV: Molnupiravir; NMV-r: nirmatrelvir–ritonavir. ^a^: Unless specified, the values are number of patients (%); ^b^: unit of measurement is × 10^9^/Liter. *: Statistical significance with *p* value < 0.05. ^c^: *p* value for comparison between the 3 groups. One-way ANOVA test and Pearson’s χ^2^ test were used for continuous and categorical variables, respectively.

**Table 2 viruses-15-00610-t002:** Effectiveness of Nirmatrelvir–ritonavir and molnupiravir in preventing COVID-19 related hospitalization, respiratory failure and mortality.

	No. of Patients (% in Group)			Anti-Viral Effectiveness (Compared with No Treatment)	
COVID-Related Outcomes	No Anti-Viral Treatment (*n* = 2387)	Nirmatrelvir- Ritonavir (*n* = 302)	Molnupiravir (*n* = 578)	Nirmatrelvir–ritonavir	Molnupiravir
Hospitalization	228 (9.6%)	15 (5.0%)	44 (7.6%)	43.9% (95% CI = −1.7–69.0), *p* = 0.06	21.8% (95% CI = −10.6–44.7), *p* = 0.16
Respiratory failure	209 (8.8%)	10 (3.3%)	40 (6.9%)	66.6% (95% CI = 25.6–85.0), *p* = 0.01 *	26.4% (95% CI = −6.3–49.1), *p* = 0.10
Severe respiratory failure	91 (3.8%)	3 (1.0%)	16 (2.8%)	77.0% (95% CI = 6.9–94.3), *p* = 0.04 *	48.2% (95% CI = 0.5–73.0), *p* = 0.05 *
Mortality	120 (5.0%)	4 (1.3%)	15 (2.6%)	62.7% (95% CI = −0.6–86.2), *p* = 0.05	58.3% (95% CI = 22.9–77.4), *p* <0.01 *

*: *p* < 0.05. Anti-viral effectiveness was calculated using the formula (1−adjusted risk ratio) × 100. The risk ratios were adjusted for age, gender, underlying respiratory diagnosis (asthma, chronic obstructive pulmonary disease or bronchiectasis) and Charlson co-morbidity index.

## Data Availability

The data presented in this study are available in the manuscript and Appendix A.

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
