# Peer review of "Real-World Study on Effectiveness of Molnupiravir and Nirmatrelvir–Ritonavir in Unvaccinated Patients with Chronic Respiratory Diseases with Confirmed SARS-CoV-2 Infection Managed in Out-Patient Setting"

_viruses, 2023, doi:10.3390/v15030610_

Round 1

Reviewer 1 Report

A territory-wide retrospective cohort study was conducted to assess the clinical efficacy of MOV and NMV-r among unvaccinated patients with chronic respiratory disease, including            asthma, COPD and bronchiectasis, who were managed in out-patient setting in designated clinics if they were clinically stable. Patients were identified from the Clinical Data Analysis and Reporting System (CDARS), an electronic healthcare database managed by the Hospital Authority, which covers 90% of healthcare services of Hong Kong. Early COVID-19 treatment regimens were offered to patients with increased risk of severe disease. To identify whether antivirals were associated with protection from COVID-19 related respiratory failure, severe respiratory failure, hospitalization and mortality, log-binomial regression was used. A total of 3267 adult patients were included in the study; of them 73.1% did not receive anti-viral, 9.2% received NMV-r and 17.7% received MOV. NMV-r and MOV were effective in preventing COVID-19 related severe respiratory failure. NMV-r have extra benefits in preventing COVID-19 related respiratory failure and hospitalization, MOV was effective in preventing in-hospital mortality.

The following points need to be addressed to improve the quality of the manuscript.

Abstract

1. Conclusions should be added, since they are a repetition of study findings.

2. Grammar error in the sentence “……and in-hospital mortality with anti-viral effectiveness of and (58.3%; 95% CI, 22.9 – 77.4, p=0.005).”

Statistical analysis

1. In the statistical analysis section is reported that baseline demographic and clinical data were compared between patients with or without antivirals with one-way ANOVA; however, in Table 1 both categorical and continuous variables have been analyzed, therefore different statistical tests should have been applied. This should be clarified both in the statistical section and in the footnote to Table 1. In addition, a column summarizing NMV-r and MOV data should be added.

2. A justification for the use of the log-binomial model instead of the Poisson or negative-binomial regression model should be reported.

Results

1. A description of the statistical analysis results included in Table 1 should also be included.

2. The Table 2 is not readable in the present form.

3. In the footnote to Table 2 is reported that the analysis was adjusted for underlying respiratory disease, but all patients included in the study had chronic respiratory disease. This should be clarified. Moreover, the effectiveness should be reported as RR, but in the footnote is reported OR.

4. Description of the study results is quite confusing and should be referred to RR or [(1-RR)x100].

5. Further information should be include if available: days from COVID-19 diagnosis/symptoms onset and start of anti-viral treatment, symptomatic status, treatment adherence (completion of the full 5-day treatment course).

Discussion

1. A discussion on the different clinical benefit between NMV-r and MOV should be implemented.

2. A contextualization of the study results compared to other published studies should be included.

Reviewer 2 Report

I commend authors for this study undertaking. However, the study is affected by several limitations.

In particular, confounding by indication bias precludes from comparing the outcomes of patients treated with MOV vs. those of patients treated with NMV-r. Amongst potential confounders, co-medications or co-morbidities are important to consider. In light of this consideration, please de-emphasize comparison between the two patient groups as much as possible.  Better statistical analysis should be performed to compare the two groups.

Why 73.1% patients did not receive antivirals? I wonder whether patients not treated are truly comparable to those treated. Also what is the relevance of the conclusions since most individuals should now be vaccinated? 

Reviewer 3 Report

In summary, authors describe clinical outcomes of COVID-19 in outpatients with underlying chronic respiratory disease (asthma, COPD or bronchiectasis), untreated or treated with oral antiviral agents against COVID-19 (molnupiravir (MOV) or nirmatrelvir-ritonovir (NMV-r)) with respect to co-primary outcomes of hospitalization, respiratory failure, severe respiratory failure and death, in adult patients who have not received COVID vaccine, in Hong Kong between February and August of 2022.  This study appears to be a follow up and a subgroup analysis of the larger study by the group reference 4.

General comments:

Novelty of the study lies in direct comparison of outcomes of the two available antiviral therapeutics in real-world environment in unvaccinated patients.  Since vaccines became available prior to oral antivirals, real-world studies of oral antiviral therapies in unvaccinated individuals are limited but are important for our understanding of antiviral efficacy.

1. Unvaccinated persons: COVID-19 vaccines became available at the end of 2020 with recommendation for priority vaccination of persons with underlying lung disease and people over 65 years of age.  It is curious that authors were able to identify large number of patients who are considered high risk but who did not receive COVID-19 vaccine.  It would be helpful for the authors to comment on the vaccination strategy in this population, to help explain presence of large number of unvaccinated individuals in this group.  By February of 2022, everyone should have been vaccinated and received at least one booster.

2. Authors state that patients were not vaccinated, but do not provide explanation as to how vaccination is tracked, and whether distinction was made between unvaccinated vs unknown vaccine status.

3. In addition, for patients at high risk for disease progression, as the population in this study, one of the first-line therapies was passive antibody therapy with one of the available monoclonal antibodies.  Was use of monoclonal antibody available in the clinical setting and at the time of this study?  Would be helpful for authors to address prior receipt of either preexposure or postexposure passive antibody therapy.

Considering large group of patient who were not vaccinated and not given oral therapy, and that more than half (53%) of those who required hospitalization – died, (vs about 25% post hospitalization mortality with NMV-r, and about 33% with MOV), authors should comment on whether these were late presentations or alternative reasons for no therapy.

4. Authors do not comment on whether any of the persons included in this study experienced prior episodes of COVID-19.  Prior immunity may affect disease progression and presence (or absence) of SARS-COV-2 antibodies would have been helpful in confirming COVID-naïve population in this study.  Would be helpful for authors to comment on whether information on prior infection was available and whether those persons were excluded from analysis.

5. There are data inconsistencies for the respiratory failure subgroup, and data table is difficult to read due to formatting errors.

Specific details:

Patient characteristics (lines 140-142): authors note that majority (73.1%) of patients did not receive antiviral therapy, and of those who did receive oral therapy almost half were given MOV over NMV-r. This is curious, since NMV-r is first line therapy, and this is a high-risk population (mean age 79.4 +/- 14 years) with underlying lung disease.  Authors acknowledge and in part addressed this in discussion section lines 251-271, however, they note provider comfort, need for renal adjustment and possible drug-drug interactions with NMV-r.  Table 1 lists only 11.1 +1.9=13% of pts who received MOV being on DOAC or anti-epileptics, and NMV-r is dose-adjusted for renal impairment, so would be helpful to understand what additional drug-drug interactions or other reasons lead providers to prescribe MOV.

Table 1:

1. Please clarify what p-value in the last column refers to.  Is it p value for NMV-r vs MOV? Or comparison of each with untreated? Statistical significance comparison groups are not clear.

2. Would recommend providing race/ethnicity of the population in Table 1.  In Discussion line 272 authors refer to this as “pure” population – would recommend adjusting this wording and clarifying patient demographics.

Table 2:

1. Table is formatted with errors and is not legible with respect to anti-viral effectiveness

2. Major comment: Respiratory failure subgroup in Table 2, S1, S2 and S3, as described in section 3.1.2 lines 174-184 has data cohort patient discrepancy.  Text of section 3.1.2 describes 172 patients with respiratory failure which is consistent with what is listed in table S3 (136 group A no antiviral, 8 group B NMV-r, 28 group C MOV), while table 2/S1/S2 list 259 patients with respiratory failure (209 group A, 10 group B, 40 group C).  It is difficult to understand outcomes of this subgroup, and unclear which data is correct.

Round 2

Reviewer 1 Report

The points raised have been adequately addressed, and now the manuscript is suitable for publication

Author Response

Thank you very much for reviewing the manuscript and we are pleased to know that the revised manuscript can address all the comments.

Reviewer 2 Report

The authors have responded to my comments satisfactorily

Author Response

Thank you very much for reviewing the manuscript and we are pleased to know that the revised manuscript can address all the comments. We have checked the English language again and several typo and spelling errors corrected.

Reviewer 3 Report

I thank the authors for careful attention to reviewer comments and clarification of data presentation, methods and conclusions. I would suggest making Table 2 looks same as S2-4.  Presentation of results in main manuscript with % is confusing and does not align with text of results in section 3.1.1.  In addition, Table 2 notes Antiviral Effectiveness of xx (MNV-r or MOV) compared with no treatment, while supplemental tables list A vs B - no-treatment group first.  Would recommend adjusting consistency for denominator being no-treatment group.  

Author Response

  • Table 2 is amended according to the advice and the number and percentage of patients that were shown in S2-4 included.
  • We amended the presentation of the results in the manuscript to include number of events/total patient number in each group and the percentage, and the presentation method from 3.1.1 to 3.1.4 is consistent.
  • S2 to S4 are amended to be consistent with the way of presentation as in Table S2.